# Surgical Treatment of Active Endocarditis Pre- and Post-COVID-19 Pandemic Onset

**DOI:** 10.3390/biomedicines12010233

**Published:** 2024-01-19

**Authors:** Elisa Mikus, Mariafrancesca Fiorentino, Diego Sangiorgi, Costanza Fiaschini, Elena Tenti, Elena Tremoli, Simone Calvi, Antonino Costantino, Alberto Tripodi, Fabio Zucchetta, Carlo Savini

**Affiliations:** 1Cardiovascular Department, Maria Cecilia Hospital, GVM Care & Research, 48031 Cotignola, Italy; francescafiorentino@hotmail.it (M.F.); dsangiorgi@gvmnet.it (D.S.); etenti@gvmnet.it (E.T.); etremoli@gvmnet.it (E.T.); scalvi@gvmnet.it (S.C.); antonino@costantinorc.com (A.C.); albertotripodi@hotmail.com (A.T.); fabiozucchetta@gmail.com (F.Z.); csavini@gvmnet.it (C.S.); 2Cardiac Surgery Department, IRCCS Azienda Ospedaliera Universitaria di Bologna, 40138 Bologna, Italy; cfiaschini@gmail.com; 3Department of Experimental Diagnostic and Surgical Medicine (DIMEC), University of Bologna, 40126 Bologna, Italy

**Keywords:** active endocarditis, COVID-19 pandemic, valve surgery, pathogens

## Abstract

Background: Despite advanced diagnosis and treatment, infective endocarditis (IE) is a potentially life-threatening condition. The impact of COVID-19 on the diagnosis and outcome of the surgical treatment of IE is uncertain. The aim of this study was to analyze the incidence, characteristics, and outcomes of surgically treated IE before and after the COVID-19 pandemic. Methods: This study retrospectively analyzed the data of 535 patients who underwent valve surgical procedures for IE between January 2010 and December 2022 in a single cardiac surgery center. Patients were divided into two groups depending on the date of their operation: before (*n* = 393) and after (*n* = 142) COVID-19 onset. In order to balance the groups, inverse probability of treatment weighting (IPTW) calculated from the propensity score (PS) was applied. Weighted univariate logistic regressions were reported for outcomes; weights were derived from IPTW. Interrupted time series analysis (ITSA) according to Linden’s method was used to evaluate the changes in the manifestation of IE after 11 March 2020. Results: Patients from the post-COVID-19 cohort (after 11 March 2020) had a greater number of comorbidities such as diabetes (29.6% vs. 16.3% *p* = 0.001), hypertension (71.1% vs. 59.5% *p* = 0.015), and preoperative kidney injury requiring dialysis (9.2% vs. 2.5% *p* = 0.002), but the median additive and logistic EuroSCORE were not statistically different. In the post-COVID-19 group, we observed a greater prevalence of *Staphylococcus aureus*-related endocarditis (24.5% vs. 15.4% *p* = 0.026), a consequent reduction in *Staphylococcus non aureus*-related endocarditis (12.2% vs. 20.1% *p* = 0.048), and a decrease in aortic valve replacements (43.0% vs. 53.9%), while the number of mitral valve replacements and repair was greater (21.1% vs. 15.0% and 6.3% vs. 4.3%, respectively). No differences were found in the two groups concerning early death, death, or relapse at 1 year after surgery. Data obtained by multivariable analysis identified preoperative renal dysfunction requiring dialysis as the only common risk factor for early mortality via stratifying by time periods in analysis. Conclusions: The incidence of surgically treated IE significantly increases after the COVID-19 pandemic with a higher incidence of mitral valve involvement with respect to the aortic valve. Although a delay in surgical timing occurred during the COVID-19 pandemic, data in terms of mortality and outcomes were largely unaffected.

## 1. Introduction

Infective endocarditis (IE) is a life-threatening disease characterized by the infection of the heart valves and the endocardium and has a high morbidity and mortality rate [1,2]. Commonly, IE affects the mitral valve, followed by the aortic, tricuspid, and pulmonary valves, but it can also involve the supporting structures. In the last few years, the epidemiology of IE has experienced substantial changes, especially in causative and drug-resistant organisms. The numbers of patients suffering from rheumatic heart disease decreased with a concomitant increase in infective endocarditis occurring in the elderly population or in patients presenting with new risk factors including intracardiac or intravenous devices, immunosuppressive conditions such as diabetes, hemodialysis, or intravenous drug use [3,4,5,6,7,8]. The impact of COVID-19 on the surgical treatment of IE is uncertain. At the beginning of the pandemic era, the incidence of hospital admissions for IE appeared to be decreasing, but later studies reported no changes in diagnosis and treatment of IE [9,10,11,12]. Since the occurrence of COVID-19 has become one of the major factors for patients with cardiovascular diseases, we have compared in this study the pre-COVID-19 versus post-COVID-19 incidence, demographics, and outcomes of surgically treated IE in a single cardiac surgery center.

Specifically, this study has taken into account (1) preoperative characteristics and outcomes of patients affected via surgically treated IE, (2) the trend of microbial aetiology before and after COVID-19 onset, and (3) factors associated with early and late mortality after surgery.

## 2. Materials and Methods

The information collected included demographic data, major baseline characteristics (age, sex, body mass index, creatinine clearance, preoperative condition, cardiovascular risk factors, functional status, and left ventricular ejection fraction (EuroSCORE II)), and the manifestations of IE according to the modified Duke criteria [13]. All data were collected and stored, together with intraoperative and short-term outcomes. Echocardiography was performed pre- and postoperatively. Clinical follow-up was performed at 30 days and 1 year, either with telephone interviews or patient visits.

### 2.1. Study Population

This is a monocentric, retrospective, single-center study involving 535 consecutive adult patients (aged > 18 years) with infective endocarditis scheduled for surgical treatment from January 2010 to December 2022. Although this study is a single-center experience, we believe that it is a good representation of the real world because our center is the only reference point for a large geographical area. No sample size calculation was performed, but all available patients were included in this study.

The study protocol was approved by the Romagna Ethics Committee on 30 June 2023 (prot.4497/2023 I.5/95). Individual informed consent was waived because of the retrospective nature of the data collected.

Data were gathered starting with clinical charts and collected in a specific registry and every possible effort has been made to reduce missing information. In our database, because missing values are referred to as “missingness” completely at random and arising directly from gaps in clinical charts, only complete cases were analyzed.

### 2.2. Surgical Procedure

The indication for cardiac surgery was established in the Heart Team according to the most recent ESC guidelines [14] for the treatment of IE, even if, at the time of surgery, these guidelines were still not published. The operation was performed with the aim of obtaining a radical debridement of the infected tissue.

All patients underwent intraoperative transesophageal echocardiography.

### 2.3. Statistical Analysis

Continuous variables were reported as median and interquartile range (IQR) and compared with the Mann–Whitney test; categorical variables were reported as absolute number and frequencies and compared with the chi-squared test or the Fisher exact test, as appropriate. In order to balance the groups, inverse probability of treatment weighting (IPTW) calculated from the propensity score (PS) was applied; observations falling outside of the common support were excluded (63 patients in the pre- and 9 in the post-COVID-19 period); no missing-data imputation was performed. Absolute standardized mean differences (ASMDs) were reported in order to assess balancing across groups; variables with an ASMD < 0.2 were considered as balanced [15]. Weighted univariate logistic regressions were reported for outcomes; weights were derived from IPTW. Interrupted time series analysis (ITSA) according to Linden’s method [16] was used for the assessment of slopes change before and after COVID-19 (11 March 2020) for both type of pathogen and type of valve. Multivariable logistic regression was also assessed via stratifying by time periods in analysis; LASSO (Least Absolute Shrinkage and Selection Operator) with leave-one-out cross validation (LOOCV) was applied for variable selection [17]; the Area Under ROC curve was reported in order to assess model discrimination. A scatter plot for percentages of hospitalized patients affected by endocarditis on overall number of hospitalizations per year was reported, along with the Spearman’s rho index for correlation between year and percentage of endocarditis. All analyses were performed with STATA 18.0 SE (StataCorp LLC); *p*-values < 0.05 were considered statistically significant.

## 3. Results

### 3.1. Baseline Characteristics

Five hundred and thirty-five patients who underwent surgery for infective endocarditis at our institution between January 2010 to December 2022 were included in this study. First, the number of endocarditis diagnoses during this time period among patients referred for surgery to our center was significantly greater in comparison to the pre-COVID-19 time period (*p* = 0.006) (Figure 1).

Three hundred and ninety-three patients (73.5%) out of the overall population with endocarditis underwent surgery before 11 March 2020 (pre-COVID-19 cohort); the remaining 142 patients (26.5%) underwent operations after this data (post-COVID-19 cohort). Preoperative patients’ characteristics are listed in Table 1. 

No significant differences between the two groups were found concerning age or sex. Patients from the post-COVID-19 cohort (after March 2020) had a greater number of comorbidities, such as diabetes (29.6% vs. 16.3% *p* = 0.001), hypertension (71.1% vs. 59.5% *p* = 0.015), and preoperative kidney injury requiring dialysis (9.2% vs. 2.5% *p* = 0.002). Nevertheless, the median additive and logistic EuroSCORE were not statistically different between the two groups, as well as the clinical presentation, with a comparable number of patients with heart failure or shock requiring an intra-aortic balloon pump or previous intubation. In both groups, the majority of patients underwent surgery during active endocarditis.

### 3.2. Microbiological and Operative Characteristics

Microbiological data showed important differences between the pre- and post-COVID-19 period, with a greater prevalence of *Staphylococcus aureus*-related endocarditis in the post-COVID-19 cohort (24.5% vs. 15.4% *p* = 0.026), and a consequent reduction in *Staphylococcus non aureus*-related endocarditis (12.2% vs. 20.1% *p* = 0.048) (Table 1), although at the interrupted time series analysis, the change in the slopes of the two periods was not statistically significant (Figure 2). On the other hand, the number of endocarditis cases related to other pathogens (*Streptococcus*, *Pseudomonas*, *Enterococcus*, Fungi, and other) remained stable across the two time periods (Table 1). Considering *Streptococcus*, a change in the slopes of the two periods at the interrupted time series analysis showed a statistically significant difference (*p* = 0.008), as the result of a slow increase in the first period and, even with a higher rate of patients affected by *Streptococcus*, a rapid decrease in the slope during the second time period (Figure 2).

Moreover, in our population, we noticed an increase from 16 to 19 days (*p* = 0.025) in the median time from endocarditis diagnosis to surgery. Table 2 reports patients’ operative characteristics.

In detail, the prevalence of native valve endocarditis was stable over time, whereas differences in the type of valve surgery were found. In fact, in the pre-COVID-19 era, a greater number of aortic valve replacements (53.9% vs. 43.0%) occurred, whereas in the post-COVID-19 period, a greater number of mitral valve replacements and repair occurred (21.1% vs. 15.0% and 6.3% vs. 4.3%, respectively). In line with these findings, the interrupted time series analysis showed a significant increase in mitral valve infective endocarditis (*p* = 0.027) in the post-COVID-19 cohort, an increase in tricuspid valve and multivalve infective endocarditis, and a reduction in aortic valve endocarditis (*p* = 0.078) (Figure 3). 

### 3.3. Postoperative Outcomes

Postoperative clinical outcomes are presented in Table 3.

The incidence of the most-common postoperative adverse events did not differ between the two groups, with a comparable percentage of rhythm disorders, neurological complications, or renal failure. The only reported differences were related to the incidence of postoperative low cardiac output syndrome requiring inotropic support that decreased from 24.7% in the pre-COVID-19 cohort to 10.6% in the post-COVID-19 cohort (*p* = 0.032), even if no differences were found in the need for mechanical circulatory support and in the incidence of respiratory failure (20.9% in the pre-COVID-19 cohort and 12.0% in the post-COVID-19 cohort (*p* = 0.018)). Moreover, there were no differences in the two groups concerning early death, death, or relapse at 1 year after surgery.

### 3.4. Risk Factor Analysis

We carried out two multivariable logistic regression models to identify risk factors for early mortality via stratifying by time period (Table 4).

The multivariable logistic regression analysis in the pre-COVID-19 population showed that age (OR, 1.08 [CI, 1.03–1.12], *p* = 0.001), heart failure (OR, 2.20 [CI, 1.05–4.64], *p* = 0.038), and dialysis (OR, 25.77 [CI, 4.84–137.20], *p* < 0.001) were independent factors associated with early mortality; the Area Under ROC curve was equal to 0.760. Instead, in the post-COVID-19 population, dialysis (OR, 5.17 [CI, 1.14–23.55], *p* = 0.034) and logistic EuroSCORE (OR, 1.05 [CI, 1.02–1.08], *p* = 0.001) were found as independent risk factors for early mortality; the Area Under ROC curve was equal to 0.815. Thus, the only common risk factor for early mortality in the two groups was preoperative renal dysfunction requiring dialysis.

## 4. Discussion

Overall, this study shows that when comparing patients with IE undergoing cardiac surgery before and after the COVID-19 pandemic, there is an increase in the prevalence of IE and, in addition, we observed a higher frequency of *Staphylococcus aureus*-related endocarditis concomitant with the reduction in *Streptococcus*. Moreover, in our population, a decrease in aortic valve surgery was registered in the post-COVID-19 patients, while the number of mitral valve replacements and repairs in the post-COVID-19 group was higher, although mitral valve repair is still underused. No differences in in-hospital death and 1-year survival in the two groups were found and preoperative renal dysfunction requiring dialysis emerged to be the only common risk factor for early mortality in the two groups.

IE remains a relatively uncommon condition, with an annual occurrence of approximately 3–10 cases per 100,000 individuals [18]. Despite advancements in both detection and treatment, results continue to exhibit unsatisfactory trends, as documented by the in-patient mortality rate of 18%, along with a 6-month mortality rate of 30% [18]. The incidence and severity of this disease remain unchanged largely driven by the evolving epidemiological profile of patients (an increasing proportion of older patients with more severe comorbidities) as well as an increasing number of patients with prosthetic- or device-related IE. In keeping with other studies, in our population, patients who underwent operations post-COVID-19 had a greater risk profile, with the presence of a greater number of comorbidities (arterial hypertension, diabetes, previous neurological disease, and preoperative acute renal injury or dialysis) [19]. Of note is the fact that the number of patients using injected drugs (7.7% vs. 5.1%) or with a permanent pacemaker (7.4% vs. 10.6%) increased after March 2020 in this study. With regard to the manifestations of IE, in agreement with recently published studies [19,20], after the COVID-19 pandemic, we found that the mitral valve involvement was significantly more frequent and a larger prevalence of *Staphylococcal aureus* infections occurred.

Of interest, our data are in accordance with the results of the EURO ENDO registry [21]. Indeed, IE more frequently affects men around 60 years of age where nosocomial, staphylococcal, and enterococcal endocarditis are more frequent, and oral streptococcal endocarditis is less frequent. In addition, mitral valve repair is still underused in IE.

Many published studies concerning infective endocarditis focus on diagnostic and therapeutic management, surgical indications and patient selection, operative risk assessment, and timing of surgery in relation to specific systemic circumstance. On the other hand, the impact of the COVID-19 pandemic on the surgical treatment of IE is still uncertain. Indeed, Novelli et al. [22] analyzed the national trends in admissions, demographics, and outcomes of IE before and after the onset of COVID-19, using a national sample of IE admissions between 2016 and 2022 from the Vizient Clinical Database. Only a small percentage of patients (14%) underwent surgery. In this subgroup, these authors observed no changes in in-hospital mortality before and after the COVID-19 pandemic. There was an increase in in-hospital stroke (7.6% vs. 8.7%, OR 1.16 [95% CI 1.04 to 1.30], *p* = 0.01) and hospital-acquired acute MI (1.1% vs. 1.5%, OR 1.33 [95% CI 1.02 to 1.73], *p* = 0.04) and decreased rates of readmission for other surgical wound complications (0.2% vs. 0.1%, OR 0.37 [95% CI 0.14 to 0.98], *p* = 0.04). The rates of all other complications remained stable. In the matched surgical cohort, length of stay from before COVID-19 to after COVID-19 was equivalent, whereas the intensive care unit days (8.5 vs. 8.0, MD −0.5 [95% CI −0.9 to 0.5], *p* = 0.04) and in-hospital mortality (7.7% vs. 6.7%, OR 0.86 [95% CI 0.74 to 0.99], *p* = 0.03) decreased. There were no differences in the rates of any complication between pre- and post-COVID-19 time periods in the surgical admissions in the matched analysis. Comparing pre- and post-COVID-19 time periods, in our patient population, the in-hospital mortality rate decreased from 12.0% to 9.9%. This lowered mortality rate highlights the importance of surgical intervention for IE, even during a global pandemic.

Pure data also show a significant decrease in postoperative respiratory failure and low cardiac output syndrome in patients undergoing surgical procedures for IE after COVID-19. Other complications remained unchanged but a progressively better trend of postoperative outcomes was observed after COVID-19, including stroke (*p* = 0.568).

IE has high morbidity and mortality rates if not interceded early in its course, and healthcare crises, such as the COVID-19 pandemic, may exacerbate delays in diagnosis and treatment [22]. Indeed, in our population, we observed a statistically significant increase in the time elapsed between diagnosis of IE and intervention, without, however, an increase in in-hospital mortality. It is possible that the introduction of a multidisciplinary “endocarditis team” and the greater attention in the diagnosis and treatment of patients suffering from IE were responsible for the good surgical outcomes, despite the delay during the first years of the COVID-19 pandemic.

## 5. Limitations

This study is limited by its retrospective nature. It is not randomized and all current data represent a single-center experience. It is also important to underline that our center admits patients suffering from endocarditis with surgical indications and not all patients are affected by this disease. In addition, a longer-term follow-up of these patients would have further strengthened to this study. 

## 6. Conclusions

In conclusion, the incidence of surgically treated IE significantly increased after the COVID-19 pandemic with a higher incidence of mitral valve involvement instead of aortic valve. Patients had a greater number of comorbidities, but a similar EuroSCORE. Despite the pandemic and the delay in surgical timing, data in terms of mortality and outcomes have improved and were largely unaffected by the COVID-19 pandemic.

## Figures and Tables

**Figure 1 biomedicines-12-00233-f001:**
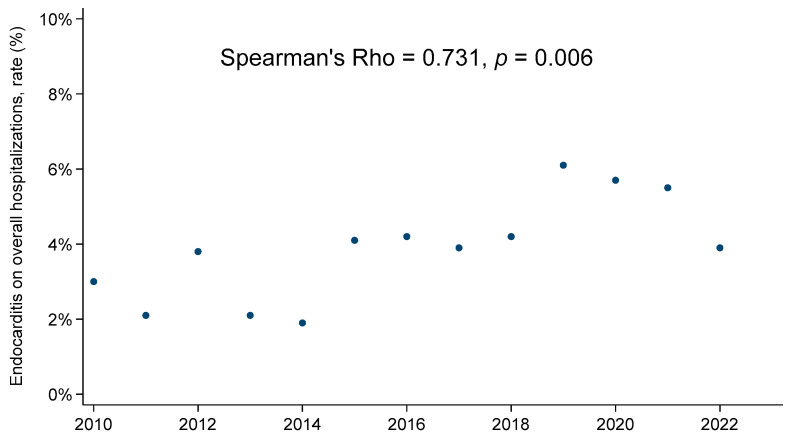
Endocarditis on overall hospitalizations rate, annual trend.

**Figure 2 biomedicines-12-00233-f002:**
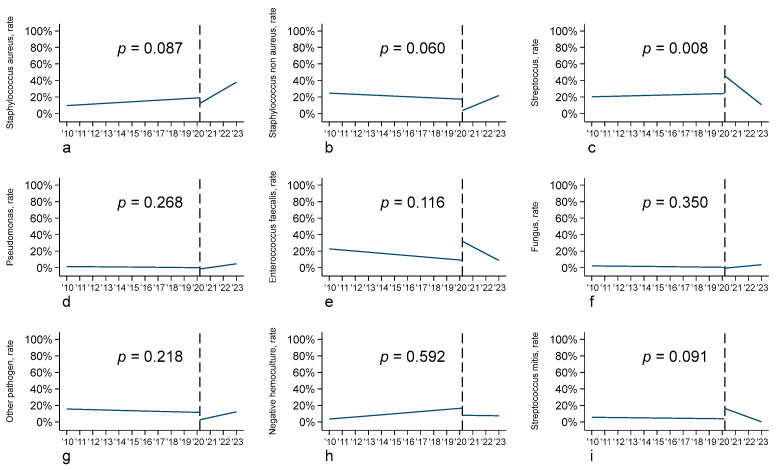
Interrupted time series analysis (ITSA) according to pathogen type: (**a**) Staphylococcus aureus; (**b**) Staphylococcus non aureus; (**c**) Streptococcus; (**d**) Pseudomonas; (**e**) Enterococcus faecalis; (**f**) fungus; (**g**) other pathogen; (**h**) negative hemoculture; (**i**) Streptococcus mitis.

**Figure 3 biomedicines-12-00233-f003:**
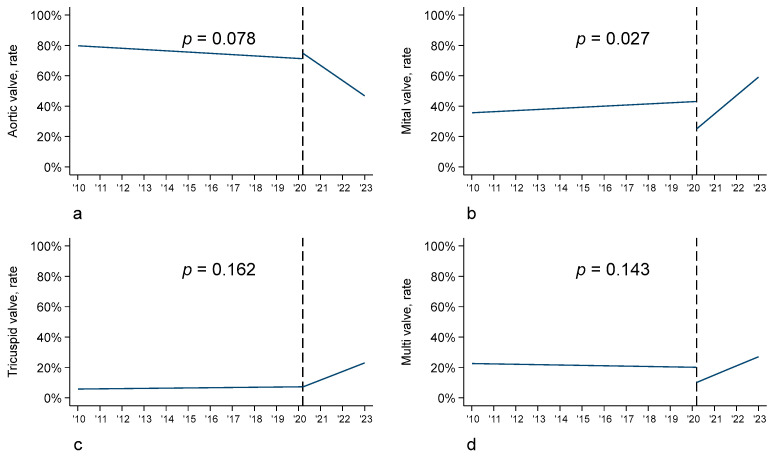
Interrupted time series analysis (ITSA) according to valve: (**a**) aortic valve; (**b**) mitral valve; (**c**) tricuspid valve; (**d**) multivalve.

**Table 1 biomedicines-12-00233-t001:** Preoperative characteristics.

	Pre-COVID-19	Post-COVID-19	*p*	ASMD	ASMD IPTW
*n*, %	393 (73.5%)	142 (26.5)			330/133
Age, median (IQR)	69.0 (57.0–76.0)	68.0 (58.0–75.0)	0.543	0.073	0.049
Female, *n* (%)	121 (30.8)	42 (29.6)	0.832	0.007	0.006
Endocarditis-intervention days, median (IQR)	16.0 (0.0–30.0)	19.0 (12.0–32.0)	0.025	0.042	0.036
*Staphylococcus aureus*, *n* (%)	53 (15.4)	34 (24.5)	0.026	0.220	0.023
*Staphylococcus non aureus*, *n* (%)	69 (20.1)	17 (12.2)	0.048	0.209	0.053
*Streptococcus*, *n* (%)	78 (22.7)	40 (28.8)	0.162	0.127	0.013
*Pseudomonas*, *n* (%)	2 (0.6)	2 (1.4)	0.327	0.083	0.015
*Enteroccoccus faecalis*, *n* (%)	49 (14.2)	29 (20.9)	0.077	0.194	0.062
*Fungus*, *n* (%)	4 (1.2)	2 (1.4)	1.000	0.020	0.004
Other pathogen, *n* (%)	45 (13.1)	10 (7.2)	0.081	0.201	0.081
Hypertension, *n* (%)	234 (59.5)	101 (71.1)	0.015	0.214	0.006
Diabetes, *n* (%)	64 (16.3)	42 (29.6)	0.001	0.318	0.009
Obesity, *n* (%)	79 (20.1)	24 (16.9)	0.457	0.089	0.093
Chronic obstructive pulmonary disease, *n* (%)	45 (11.5)	11 (7.7)	0.264	0.147	0.095
Ejection fraction %, median (IQR)	57.0 (50.0–61.0)	60.0 (54.0–65.0)	0.150	0.149	0.032
Drug addict, *n* (%)	20 (5.1)	11 (7.7)	0.294	0.060	0.028
Reoperation, *n* (%)	183 (46.6)	61 (43.0)	0.492	0.077	0.025
Previous intra-aortic balloon pump, *n* (%)	3 (0.8)	0 (0.0)	0.569	0.135	0.158
Peripheral arterial disease, *n* (%)	49 (12.5)	18 (12.7)	1.000	0.009	0.083
Neoplasm, *n* (%)	48 (12.2)	9 (6.3)	0.057	0.202	0.051
Previous neurological disease, *n* (%)	74 (18.8)	38 (26.8)	0.054	0.167	0.001
Unstable angina, *n* (%)	2 (0.5)	1 (0.7)	1.000	0.059	0.003
Shock, *n* (%)	16 (4.1)	9 (6.3)	0.352	0.147	0.007
Heart failure, *n* (%)	108 (27.5)	32 (22.5)	0.267	0.111	0.025
Myocardial infarction 90 days, *n* (%)	7 (1.8)	6 (4.2)	0.117	0.190	0.002
Previous intubation, *n* (%)	25 (6.4)	11 (7.7)	0.562	0.073	0.017
Pulmonary hypertension, *n* (%)	26 (6.6)	11 (7.7)	0.700	0.087	0.110
Additive EuroSCORE, median (IQR)	10.0 (7.0–13.0)	10.0 (7.0–12.0)	0.690	0.042	0.021
Logistics EuroSCORE, median (IQR)	19.4 (8.0–42.1)	19.6 (7.9–35.0)	0.617	0.107	0.020
Active endocarditis, *n* (%)	335 (85.2)	130 (91.5)	0.060	0.121	0.016
Cirrhosis, *n* (%)	4 (1.0)	0 (0.0)	0.578	0.110	0.084
Chronic kidney disease (creatinine > 2.0 mg/dl), *n* (%)	44 (11.2)	18.5 (13.0)	0.264	0.057	0.112
Dialysis, *n* (%)	10 (2.5)	13 (9.2)	0.002	0.281	0.018
Permanent pacemaker, *n* (%)	29 (7.4)	15 (10.6)	0.284	0.091	0.035
Abscess, *n* (%)	105 (26.7)	44 (31.0)	0.328	0.111	0.054
Vegetation, *n* (%)	318 (80.9)	123 (86.6)	0.157	0.094	0.087
Leaflet perforation, *n* (%)	47 (12.0)	33 (23.2)	0.002	0.317	0.045
Prosthesis detachment, *n* (%)	78 (19.8)	16 (11.3)	0.021	0.216	0.029

**Table 2 biomedicines-12-00233-t002:** Operative characteristics.

	Pre-COVID-19	Post-COVID-19	*p*
*n*	393	142	
Type of valve surgery:			<0.001
- Aortic valve replacement, *n* (%)	212 (53.9)	61 (43.0)	
- Mitral valve replacement, *n* (%)	59 (15.0)	30 (21.1)	
- Mitral valve repair, *n* (%)	17 (4.3)	9 (6.3)	
- Tricuspid valve replacement, *n* (%)	5 (1.3)	4 (2.8)	
- Tricuspid valve repair, *n* (%)	5 (1.3)	5 (3.5)	
- Mitral and aortic valve repair or replacement, *n* (%)	69 (17.6)	13 (9.2)	
- Other, *n* (%)	26 (6.6)	20 (14.1)	
Native valve endocarditis, *n* (%)	241 (61.3)	94 (66.2)	0.314

**Table 3 biomedicines-12-00233-t003:** Outcomes.

	Pre-COVID-19	Post-COVID-19	Weighted OR	95% CI		*p*
*n*	393	142				
Infection sepsis, *n* (%)	37 (9.4)	9 (6.3)	0.521	0.201	1.349	0.179
Multiorgan failure *n* (%)	32 (8.1)	3 (2.1)	0.294	0.083	1.045	0.058
Permanent pacemaker implantation, *n* (%)	40 (10.2)	11 (7.7)	0.591	0.261	1.338	0.207
Cardiocirculatory arrest, *n* (%)	10 (2.5)	2 (1.4)	0.774	0.134	4.466	0.775
Atrial fibrillation, *n* (%)	111 (28.2)	32 (22.5)	0.635	0.364	1.108	0.110
Low cardiac output, *n* (%)	97 (24.7)	15 (10.6)	0.452	0.218	0.936	0.032
Intra-aortic balloon pump, *n* (%)	20 (5.1)	6 (4.2)	0.836	0.259	2.700	0.765
Extracorporeal membrane Oxygenation, *n* (%)	3 (0.8)	0 (0.0)	/			
Respiratory failure, *n* (%)	82 (20.9)	17 (12.0)	0.431	0.215	0.864	0.018
Stroke, *n* (%)	24 (6.1)	4 (2.8)	0.568	0.146	2.205	0.413
Transient ischemic attack, *n* (%)	5 (1.3)	2 (1.4)	2.240	0.396	12.677	0.362
Acute kidney injury, *n* (%)	75 (19.1)	16 (11.3)	0.656	0.330	1.304	0.229
Continuous venovenous Hemofiltration, *n* (%)	41 (10.4)	7 (4.9)	0.495	0.198	1.235	0.132
Major bleeding, *n* (%)	20 (5.1)	9 (6.3)	1.147	0.428	3.078	0.785
Reoperation for bleeding, *n* (%)	39 (9.9)	11 (7.7)	0.702	0.304	1.619	0.407
Wound deheiscence, *n* (%)	3 (0.8)	3 (2.1)	4.059	0.656	25.123	0.132
Early death, *n* (%)	47 (12.0)	14 (9.9)	0.751	0.359	1.572	0.448
Death at 1 year, *n* (%)	72 (18.3)	26 (18.3)	0.900	0.501	1.616	0.724
Relapse at 1 year, *n* (%)	20 (5.1)	2 (1.4)	0.286	0.065	1.270	0.100

**Table 4 biomedicines-12-00233-t004:** Multivariable logistic regressions for early death, pre- and post-COVID-19 pandemic.

	Pre-COVID-19				Post-COVID-19			
	OR	95% CI		*p*	OR	95% CI		*p*
Age	1.076	1.030	1.124	0.001	/	/	/	/
Heart failure	2.201	1.045	4.636	0.038	/	/	/	/
Dialysis	25.772	4.841	137.203	<0.001	5.172	1.136	23.554	0.034
Logistic EuroSCORE	/	/	/	/	1.046	1.018	1.075	0.001

## Data Availability

The data presented in this study are available on request from the corresponding author. The data are not publicly available due to data protection directive 95/46/EC.

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
