# Peer review of "Surgical Treatment of Active Endocarditis Pre- and Post-COVID-19 Pandemic Onset"

_biomedicines, 2024, doi:10.3390/biomedicines12010233_

Round 1

Reviewer 1 Report

Comments and Suggestions for Authors

Dear authors: I read your manuscript with great interest. I appreciate your efforts to explore the complex topic of COVID-19 and the incidence and consequences of endocarditis. However, please allow me to address a few points.

Currently, your manuscript is more of an epidemiological study, but your patient cohort should give a more accurate insight into the situation. The term "post-COVID" is puzzling. Your cohort spanned from January 2010 to December 2022, and you divided the groups into a pre-COVID and a post-COVID group with a cut-off of March 2020. The latter group would be better referred to as the "during-COVID" group or simply COVID group. Furthermore, it is at least methodologically questionable to compare a 10-year period with all its developments during such a long period with a two-year period during a pandemic.

Further suggestions for clarification:

2. how many patients were vaccinated? 

3. how often were they vaccinated?

4. what type of vaccination did they receive? 

5. how many patients were diagnosed with COVID-19 infection?

6. how many patients had an active COVID-19 infection?

7. how many patients were found to have residual COVID-19 lung infection on CT scan?

You mentioned a delay in surgical timing. However, there is no data to support this. Therefore the following suggestions:

8. how long was the time between disease onset and diagnosis before and during the COVID pandemic?

9. how long was the period between diagnosis and treatment before and during COVID-19?

10. was there a difference in the percentage of surgical versus conservative treatment before and during the pandemic?

Comments on the Quality of English Language

quite ok, I'm not a native speaker either

Author Response

Response to Reviewer#1

Dear authors: I read your manuscript with great interest. I appreciate your efforts to explore the complex topic of COVID-19 and the incidence and consequences of endocarditis. However, please allow me to address a few points.

Currently, your manuscript is more of an epidemiological study, but your patient cohort should give a more accurate insight into the situation. The term "post-COVID" is puzzling. Your cohort spanned from January 2010 to December 2022, and you divided the groups into a pre-COVID and a post-COVID group with a cut-off of March 2020. The latter group would be better referred to as the "during-COVID" group or simply COVID group. Furthermore, it is at least methodologically questionable to compare a 10-year period with all its developments during such a long period with a two-year period during a pandemic. Further suggestions for clarification:

COMMENT 1: how many patients were vaccinated? how often were they vaccinated? what type of vaccination did they receive? how many patients were diagnosed with COVID-19 infection? how many patients had an active COVID-19 infection? how many patients were found to have residual COVID-19 lung infection on CT scan?.

ANSWER 1: We thank the reviewer for his observation. Unfortunately we are unable to give these information for a number of reasons. First of all, because patients were recruited in different time periods, both before the availability of vaccines as well as after. In addition, some patients require urgent surgical intervention. Thus this type of information is not consistent in our study group.

For this reason we have decided to address the question considering COVID 19 infection as an incidental event (using an Interrupted Time Series Analysis commonly adopted in public health to evaluate the impact of interventions or exposures) and comparing patients positive or negative with respect to it.

COMMENT 2:  You mentioned a delay in surgical timing. However, there is no data to support this. Therefore the following suggestions:

how long was the time between disease onset and diagnosis before and during the COVID pandemic?

ANSWER 2: We thank the reviewer for the observation. In the results section in table 1 line 1 there is the time between disease onset and diagnoses. In addition in the text  we write: “Moreover, in our population we noticed an increase in the median time from endo-carditis diagnosis to surgery, that went from 16 to 19 days (p=0.025).”

COMMENT 3:  was there a difference in the percentage of surgical versus conservative treatment before and during the pandemic?

ANSWER 3: We thank the reviewer for the pertinent observation. I am very sorry, but I am not able to answer this question: Maria Cecilia Hospital is the reference cardiac surgery center in the “Area Vasta Romagna", where, however, patients with cardiac surgery indications belong, but not all those suffering from endocarditis

MODIFIED TEXT: Limitation “It is also important to underline that our Center admits patients suffering from endocarditis with surgical indications and not all those affected by this disease”.

Reviewer 2 Report

Comments and Suggestions for Authors 1. This report summarized in detail how comorbid conditions such as diabetes, hypertension and kidney disease as well as the microbial diagnosis and location of endocarditis evolved with the covid-pandemic. The outcome (early & 1-year did not change) 2. Manuscript's strengths and weaknesses: - strength: thorough statistical analysis (propensity score, IPTW) - weakness: scoring system: Euroscore II seems more adequate compared to additive or logistic ES; even so, a specific score for IE should be recommended
3. There is one major point: the use of a specific endocarditis related score. 4. Minor for the improvement of the manuscript: Language editing

The spelling of Staphylococcus should be checked throughout. Species should be written in italics.

Repetition (line 158-166) should be avoided 

Figure 3: add the pathogens as a legend 

With 14 early deaths, entering 2 predictors for the post-Covid era in table 4 should be interpreted cautiously (overfitting)  

Why were not more endocarditis specific risk scores used (Palsuse, DeFeo, Ancla, Risk-E, EndoSCORE, Meld-XI, Costa,...) for endocarditis? 

Comments on the Quality of English Language

The use of language seems OK. 

Author Response

Response to Reviewer#2

  1. This report summarized in detail how comorbid conditions such as diabetes, hypertension and kidney disease as well as the microbial diagnosis and location of endocarditis evolved with the covid-pandemic. The outcome (early & 1-year did not change)
  2. Manuscript's strengths and weaknesses: - strength: thorough statistical analysis (propensity score, IPTW) - weakness: scoring system: Euroscore II seems more adequate compared to additive or logistic ES; even so, a specific score for IE should be recommended
  3. There is one major point: the use of a specific endocarditis related score.
  4. Minor for the improvement of the manuscript: Language editing

COMMENT 1:  The spelling of Staphylococcus should be checked throughout. Species should be written in italics.

ANSWER 1: We thank the reviewer for his observation. We have checked and modified the text, including the graphical abstract, based on your observation.

MODIFIED TEXT: Graphical abstract e all text

COMMENT 2:  Repetition (line 158-166) should be avoided

ANSWER 2: We thank the reviewer for the observation. I apologize if I didn't understand the question well. In the previous lines we talk about the percentages of pathogens in the two periods, while subsequently they are the results of the ITSA analysis, which are two different analyses

“Microbiological data showed important differences between the pre- and the post- COVID-19 period, with a greater prevalence of Staphylococcus Aureus related endocarditis in the post COVID cohort (24.5% vs 15.4% p=0.026), and a consequent reduction in Staphylococcus non Aureus related endocarditis (12.2% vs 20.1% p=0.048) (Table 1), although at the interrupted time series analysis the change in the slopes of the two periods was not statistically significant (Figure 2). On the other hand, the number of endocarditis related to other pathogens (Streptococcus, Pseudomonas, Enterococcus, Fungi and other) remained stable across the two periods of time (Table 1). Considering Streptococcus, a change in the slopes of the two periods at the interrupted time series analysis showed a statistical significant difference (p=0.008), as the result of a slowly increase in the first period and, even if with a higher rate of patients affected by Streptococcus, a rapidly decrease in the slope during the second period of time (Figure 2).”

COMMENT 3:  Figure 3: add the pathogens as a legend

ANSWER 3: We thank the reviewer for the pertinent observation.

MODIFIED TEXT: Figure 2 and 3 as follow

Figure 2. Interrupted time series analysis (ITSA) according to pathogen: Staphylococcus aureus (panel a); Staphylococcus non aureus (panel b); Streptoccus (panel c); Pseudomonas (panel d); Enteroccoccus faecalis (panel e); Fungus (panel f); Other pathogen (panel g); Negative emocolture (panel h); Streptococcus mitis (panel i)

Figure 3. Interrupted time series analysis (ITSA) according to valve: Aortic valve (panel a); Mitral valve (panel b); Tricuspid valve (panel c); Multi valve (panel d)

COMMENT 4:  With 14 early deaths, entering 2 predictors for the post-Covid era in table 4 should be interpreted cautiously (overfitting) 

ANSWER 4: We thank the reviewer for the pertinent observation. We agree that, as a general rule of thumb, according to Long and Freese (Regression Models for Categorical Dependent Variables Using Stata, 2006) 10 outcomes for each covariate are needed in order to avoid overfitting; despite that, LOOCV approach guarantees an optimal external validation; moreover, for dialysis we have almost identical results at the univariate analysis: OR=5.156 (1.346-19.754), p=0.017.

COMMENT 5:  Why were not more endocarditis specific risk scores used (Palsuse, DeFeo, Ancla, Risk-E, EndoSCORE, Meld-XI, Costa,...) for endocarditis?

ANSWER 5: We thank the reviewer for the pertinent observation. As you mentioned in your question, many available IE risk-scores exist. However, as recently published by Rizzo et al. (Rizzo V, Salmasi MY, Sabetai M, Primus C, Sandoe J, Lewis M, Woldman S, Athanasiou T. Infective endocarditis: Do we have an effective risk score model? A systematic review. Front Cardiovasc Med. 2023 Feb 20;10:1093363. doi: 10.3389/fcvm.2023.1093363. eCollection 2023.PMID: 36891243), despite the variety of available scores, their development has been limited by small sample size, retrospective collection of data, with lack of external validation, limiting their transportability; in particular, only 4 scores on in-hospital mortality were externally validated (Risk-E Endocarditis Score, APORTEI score, AEPEI Score II, LOPEZ); nonetheless, these scores used a stepwise selection approach for variable selection, which is a now obsolete technique whose limits have been widely demonstrated.

Round 2

Reviewer 1 Report

Comments and Suggestions for Authors

Dear Authors: Thank you for your response.